# Fast Deployment of a UWB-Based IPS for Emergency Response Operations

**DOI:** 10.3390/s23094193

**Published:** 2023-04-22

**Authors:** Toni Adame, Julia Igual, Marisa Catalan

**Affiliations:** IoT Research Group, Fundació i2CAT, Gran Capità 2-4, 08034 Barcelona, Spain

**Keywords:** IPS, UWB, IEEE 802.15.4a, GNSS-RTK, disaster management, first responders, emergency response

## Abstract

A wide range of applications from multiple sectors already use ultra-wideband (UWB) technology to locate and track assets precisely. This is not the case, however, for first responder localization during emergency response (ER) operations, which are highly conditioned by procedural and environmental constraints. After analyzing these limitations and reviewing the current state-of-the-art solutions, this work presents a UWB-based indoor positioning system (IPS) that relies on the global navigation satellite system real-time kinematic (GNSS-RTK) technology to quickly, accurately, and safely deploy its required infrastructure on site. A set of tests conducted on a two-story building prove the suitability of such a system, providing an average accuracy of less than 1 meter for static targets and the ability to faithfully reproduce the path followed by a mobile target inside the building. The obtained results strengthen the presented approach and pave the way for more sophisticated UWB-based IPSs that would include unmanned aerial vehicles (UAVs) and/or mobile robots to speed up network deployment even more while offering additional ER services.

## 1. Introduction

Modern societies are exposed to multiple and heterogeneous risks, such as natural disasters, health-related threats, attacks on public spaces, industrial hazards, and household accidents. Some of these threats have become even more prevalent in recent years, where the combined impacts of climate-related disasters and COVID-19 have directly affected hundreds of millions of people around the world [1].

Disaster management systems provide emergency intervention teams with tools to effectively prepare for, respond to, and recover from any of the aforementioned situations. When being part of such systems, mobile technologies, computer applications, and digital data are able to speed up and increase the effectiveness of the mobilization, preliminary situation assessment, and intervention in disaster-affected areas [2,3].

Particularly, the Internet of Things (IoT) paradigm has long proven its suitability in emergency response (ER) operations, thanks to its ability to collect data in real time, from heterogeneous sources, and at a fine granular level [4,5,6]. These data are later transmitted wirelessly, thus delivering rich information in a continuous flow to the command and control center. Consequently, IoT technologies may help boost cooperation between organizations, improve situational awareness, and enable complete visibility of the response force and the remaining resources in ER operations [7,8,9].

One of the most important agents in the ER ecosystem are first responders (FRs), who are among the first to arrive to the scene of an emergency and provide assistance or incident resolution. While it is true that some existing IoT-based tools already enhance the operational capacity and increase the safety of FRs in the field [10,11,12,13,14], there still exist capability gaps, such as “the ability to know the location of FRs and their proximity to risks and hazards in real time”, as pointed out by the International Forum to Advance First Responder Innovation (IFAFRI) [15].

To shed more light on this issue, this paper describes the implementation of an indoor positioning system (IPS) based on the ultra-wideband (UWB) technology without pre-existing infrastructure and assesses its performance in a real environment. The main contribution of the current work consists in using GNSS-RTK technology to quickly and accurately self-locate IPS reference devices outside of the target building, which speeds up on-field deployment and therefore makes the whole system suitable for potentially urgent interventions due to its *plug-and-play* approach.

The remainder of this document is organized as follows: Section 2 reviews the current IPS systems and technologies, with a special focus on those based on UWB. Section 3 describes the proposed UWB-based IPS and Section 4 elaborates on the system’s implementation and the evaluation testbed. Section 5 provides information on the obtained results and, lastly, Section 6 compiles the lessons learned and presents the future work.

## 2. State of the Art

The goal of this section is to review the state of the art on the main topics related to IPSs in general, the particular requirements and constraints of positioning in ER operations, and the main features that make UWB a suitable technology for such activities.

### 2.1. Indoor Positioning Systems (IPSs)

Since its release for civilian use in the 1990s, GPS’s indisputable success in outdoor positioning and navigation systems has been mainly substantiated by its global availability and cost-effectiveness. Nonetheless, degradation or even unavailability of satellite signals in indoor scenarios make GPS unsuitable for the stringent accuracy requirements of fine localization. To be more specific, the typical accuracy for a smartphone’s GPS receptor ranges from 5 to 10 meters depending on the environmental conditions, whereas most indoor applications preferably require mean accuracies around 1 to 2 m [16,17].

The need for filling the gap left by GPS resulted in the emergence of the IPS concept, which encompasses any automatic system responsible for identifying, locating, and tracking people and objects within a pre-defined area on a real-time basis. Depending on the kind of technology and density of the deployed sensing devices, an IPS may provide different ways to express position and accuracy levels [18]:**Absolute position:** The coordinates (and altitude) of the tracked object. The accuracy is measured quantitatively; for instance, from *meter-level* to *centimeter-level*.**Relative position:** The 2D or 3D distance to a fixed point. The accuracy is also expressed quantitatively.**Symbolic position:** It implies the presence of the item in a specific area (for instance, *zone-level* places it on the correct floor or within a pre-determined subarea, while *room-level* does so in a specific room) or near something or someone.

The current study only considers the use of an IPS to estimate the exact position of a person/object. It leaves out of the scope the usage of an IPS for proximity detection (e.g., *location at choke points*), where the outcome is to determine *what specific location you are near*.

A large variety of IPS technologies has been proposed over the years, but none of them has achieved high prevalence over the rest [19]. Such is the heterogeneity of these technologies that they can be classified into the following fields: infrared, ultrasound, audible sound, magnetic field, optical and vision, radio frequency (RF), visible light, dead reckoning, and hybrid [20]. Figure 1 compares two of the most important features (range and accuracy) of the main positioning technologies.

Infrared, ultrasound, optical and vision, and visible light technologies can only provide positioning estimations in line-of-sight (LOS) conditions; that is, when no obstacles or walls stand between the IPS infrastructure and the device or person to locate. This is particularly limiting in ER operations, as it would entail the installation and calibration of such a great amount of devices inside the target indoor environments, which, in addition, usually have poor visibility and acoustic conditions.

Conversely, RF systems perform better in such environments, due to their ability to operate in non-line-of-sight (NLOS) conditions. In particular, most of the proposed wireless technologies to build an IPS over them are short-range (e.g., RFID, BLE, UWB, WiFi, and ZigBee) (see Table 1 for further details), because emerging long-range IoT technologies (e.g., Sigfox, LoRa, IEEE 802.11ah, and Weightless, among others) report a low accuracy, especially in indoor environments [22]. Regardless, the unpredictable propagation of wireless signals and the associated delay deterioration in the presence of obstacles and/or interference have typically represented the main challenges for all these technologies [23].

Dead-reckoning systems are based on the use of an inertial measurement unit (IMU) to estimate the target’s position in real time. An IMU usually contains accelerometers, gyroscopes, and magnetometers, being able to determine an object’s current position by only considering its past position and current speed and direction and altogether conforming to a *building-independent* IPS [25]. Though widely employed, the main drawback of these systems is the fast-growing drift over time when operating alone, which requires either periodic recalibrations or combination with other technologies.

Lastly, NASA has pioneered the use of probes to generate quasi-static magnetic fields that are read with a sensor placed outside the target scenario [26]. Unlike radio waves, these magnetic fields are not disturbed by metallic or other structures commonly found in building environments. Hence, this technology would in theory deliver higher localization accuracies once it evolves from its current early stage and becomes a reality.

All in all, the rest of the current section will elaborate on RF-based technologies for IPSs due to their higher suitability for ER operations, maturity level, availability, and effectiveness.

### 2.2. Parameter-Based Positioning

Regardless of the underlying RF technology, the estimation of the position of an object in a given time relies on the physical parameters of the signal acquired using the IPS. These parameters can be classified into [27]:**Distance-based:** related to the observed distance (*ranging*) between a reference device and the object to locate.−**Signal-based:** signal properties, such as the received signal strength indicator (RSSI) and/or the channel state information (CSI), may be used in distance estimation.−**Time-based:** the propagation time of a wireless signal between a transmitter and a receiver allows one to compute the existing distance between both devices. Well-known parameters to finally obtain this distance are the time of arrival (ToA), the time of flight (ToF), the round trip time (RTT), the two-way ranging (TWR), and the time difference of arrival (TDoA).−**Phase-based:** both the phase of arrival (PoA) and the phase difference of arrival (PDoA) estimate the distance by measuring the phase of the carrier signal.**Direction-based:** related to the direction of transmitted radio waves. Typical parameters on this matter are the angle of arrival (AoA), the angle difference of arrival (ADoA), and the direction of arrival (DoA).

Once the IPS has collected enough values of the localization parameters among the different network devices, they are introduced into a positioning technique that will produce as an output a new estimated position. Again, the existing techniques can be classified into different groups [28]:**Lateration techniques:** they use the multiple obtained rangings between the reference devices and the target object as well as the well-known positions of reference devices.**Angulation techniques:** in this case, the obtained angles between the reference devices and the target object are employed in combination with the well-known positions of reference devices.**Scene analysis/Fingerprinting:** a pre-recorded mapping, consisting of training data, is compared to a new *fingerprint* by using similarity measurement, statistical, or machine learning methods to identify the best match as the location of the target object [29].**Proximity:** the target position is assumed to be that of the closest reference device.

### 2.3. IPSs for ER Operations

Delivering an accurate (∼1 m) and sustainable-over-time (>tens of minutes) real-time localization for ER operations in indoor environments is a long-sought technical feature [30] with a great potential to increase the efficiency and safety of ER operations such as firefighter interventions. Positioning information and walked paths of FRs offer a higher context awareness and a better common operational picture, which would allow an incident commander to evacuate or rescue at-risk or trapped FRs, identify personnel at key locations, notify FRs if they are in proximity to threats, and design escape routes [15].

Unlike other IPS use cases, ER operations are highly conditioned by procedural and environmental constraints. On the first hand, any rescue mission has to cope with fully unpredictable real-life scenarios while following strict and hierarchical procedures. A suitable IPS should therefore adapt to existing ER processes and equipment, leaving little or no room for complex infrastructure deployments, system configurations, or positioning calibrations, which can become a burden during high-stress situations.

On the other hand, ER environments are typically *unstructured* or *unknown*, so that a running IPS has no prior information about the structure or the signal propagation conditions of the target building [31]. Moreover, hazardous working conditions (involving fire, floods, high temperature, darkness, or even a collapse) may affect not only a building’s structure and power supply (hindering the operation of pre-existing IPS infrastructure), but also the security of FRs themselves and proper operation of the equipped electronic devices.

Some commercial products already provide fine indoor positioning, but they either require pre-installation/calibration of reference devices in the building or the use of pre-existent building maps, which notably complicates their applicability in ER environments [31]. IPS market penetration is indeed much more consolidated in other activity sectors such as healthcare, industry, retail, hospitality, and transportation [32,33,34]. For its part, the scientific literature has also extensively addressed the issue of suiting current IPSs to the harsh conditions commonly found in ER operations. The result is a plethora of systems, with those based on RF, IMUs, and hybrid approaches predominating [30,31,35,36,37,38,39,40,41,42].

### 2.4. Ultra-Wideband (UWB)

IEEE 802.15.4a appeared in 2007 as an amendment to the IEEE 802.15.4-2006 standard for low-rate wireless personal area networks (LR-WPANs) [43]. Two additional physical (PHY) layers were defined to support higher throughput, longer ranges, lower power consumption, and precision ranging capabilities: chirp spread spectrum (CSS) and ultra-wideband (UWB) [44].

UWB uses a high-rate pulse-repetition scheme consisting of very short pulses (typically less than 2 ns wide, with an instantaneous bandwidth greater than 500 MHz), which results in an extremely high time resolution of signals even in environments with multipath propagation. This feature allows one to determine the ToF (and, consequently, the distance) between two devices at the centimeter level as long as they are in LOS conditions [19,45]. Conversely, signal-based ranging-estimation techniques such as RSSI deliver meter-level accuracy. Despite being widely employed in technologies such as WiFi and Bluetooth, RSSI records are more prone to instability caused by interference and fading [46].

For its part, UWB is rarely affected by interference from other communication devices or external noise except for the unlicensed use of the 6 GHz band (e.g., in WiFi 6), which shares some of its channels [47]. Lastly, the particular wavelength of UWB signals holds good penetration capabilities for most common building materials such as concrete, glass, and wood, thus enabling positioning even in NLOS conditions. Altogether, this makes UWB an eligible technology for IPSs operating in heterogeneous sectors, even those involving fast-moving objects, with fine localization requirements both in accuracy and time.

### 2.5. UWB-Based Positioning Systems

The harmonized European standard “ETSI EN 302 065-1 Part 2” [48] describes the electromagnetic requirements for UWB location-tracking devices. (Similarly, the FCC has also introduced its own electromagnetic limitations for UWB valid across the United States [49].) UWB equipment is categorized there according to the operational frequency range and the type of tracking it performs. In this context, three different systems are defined:LT1 Systems: General location tracking of people and objects in the 6 GHz to 9 GHz region.LT2 Systems: Person and object tracking and industrial applications at well-defined locations in the 3.1 GHz to 4.8 GHz region.LAES Systems: Staff tracking in location-tracking applications for emergency and disaster situations (LAES) in the 3.1 GHz to 4.8 GHz region. Licences may be required.

Some power and time transmission limitations that currently apply to LT2 systems are somehow alleviated in LAES systems, due to the highly critical and transient character of operations conducted at the scene of an emergency:The maximum allowed value of the mean and peak effective isotropic radiated power (e.i.r.p.) is 20 dBm higher in LAES systems in the band defined between 3.4 and 4.2 GHz.The duty cycle of both LT2 and LAES systems is limited to a maximum of 5% per second. However, a duty cycle limited to a maximum of 1.5% per minute and a maximum $T_{on}$ duration of 25 ms may also apply to LT2 transmitters.

While it is assumed that any IPS working under LAES regulations should perform better than in LT1 or LT2 systems, to the best of our knowledge there are no commercial or scientific studies that corroborate this premise. In fact, the single scientific work specifically addressing LAES systems elaborates on the design of their first silicon integrated power amplifier [50].

Regardless of the operational frequency range, most UWB-based positioning systems determine the position of an object (from now on called *tag*) through a two-step approach. First, a set of rangings between the tag and a group of devices with well-known positions (from now on called *anchors*) are obtained from the analysis of the ToA, the TDoA, the TWR, and/or the PDoA. Second, the position of the tag is estimated using a geometric multilateration or a fingerprinting technique [51]. Alternatively, the angles obtained from the analysis of the AoA can be used in combination with a multiangulation tecnique.

Different entities of the IPS can run the localization algorithm: an external cloud, the fixed anchors, or the tag itself. While this task has been traditionally delegated to the cloud due to its greater computing capabilities, the growing performance of modern low-power microcontrollers allows it to be performed in the tag itself. One key advantage of this option is the latency reduction for a mobile tag to obtain its position, thus providing timely trajectory tracking [52].

## 3. System Design

This section describes the proposed UWB-based IPS that, according to the taxonomy of IPSs for emergency responders presented in [31], can be classified as a *radio-signal-based system* that relies on a *strategic deployment* with an *ad-hoc localization* principle based on a *centralized* algorithm that can be used both in *indoor* and *outdoor* environments.

### 3.1. Architecture

The architecture of the proposed UWB-based IPS is split into three main subsystems, as can be observed in Figure 2:The **GNSS-RTK self-location system** enables the fast deployment of UWB anchors outside the target building. The highly accurate self-location of these devices is conducted thanks to the information received from both GNSS satellites and GNSS-RTK ground-based reference stations.The **UWB positioning system** allows us to track an FR inside a building in real time during an ER operation. As in most RF-based IPSs, it consists of a group of anchors and a tag, which in our case will be attached to an FR.The **command and control center** includes all the necessary systems to manage the IPS configuration, store the generated positioning data, and display it remotely.

Under this architecture the anchors play a triple role: First they are used to establish an absolute and very accurate reference system for the IPS by means of the GNSS-RTK self-location system. Then, they send the obtained GNSS-RTK positions to the command and control center. Lastly, their responses to the UWB requests of the tag will be used by the latter to estimate its own position.

As for the tag, it first communicates with the data management server to obtain the well-known positions of the network anchors. Then, it feeds a multilateration algorithm with the dictionary of anchors’ positions and the UWB computed rangings to generate new position estimations as the outcome. The obtained positions will periodically be sent to the data management server.

### 3.2. Anchor Self-Location with GNSS-RTK

The global navigation satellite system (GNSS) [53] provides continuous geo-spatial positioning over the Earth. Its most-known satellite-based radionavigation system is GPS, though it also includes other systems such as Glonass, Galileo, and Beidu. The arrangement of satellite constellations guarantees the visibility of at least four satellites from any point on the planet. In this way, GNSS receivers can solve the navigation equations, obtain their coordinates (latitude, longitude, altitude), and provide accurate velocity and time. The accuracy of standard GNSS positioning ranges from a few meters to 20 m, making it infeasible for systems where accuracy is essential.

However, there are augmentation methods [54], such as differential GNSS (DGNSS) or real-time kinematic (RTK), that improve the navigation performance and enhance the accuracy. These techniques use ground-based reference stations to broadcast differential information to the GNSS unit, also called the *rover*. This information aims to cancel or reduce GNSS signal-processing errors such as satellite clock bias and satellite orbital error, as well as ionospheric and tropospheric delay, which are correlated between two nearby GNSS units.

DGNSS broadcasts as corrections the range error of the measured pseudoranges (i.e., the distance from the GNSS unit and each satellite) based on pseudo-code measurements and provides accuracy on the order of 1 m. On the other hand, RTK uses code and carrier phase measurements, which have a higher frequency and smaller noise, thus reaching centimeter-level accuracy. However, processing of the carrier measurements is subject to the carrier-phase ambiguity, that is, an unknown integer number of wavelength cycles. Still, the integer ambiguity problem can be fixed as long as dual-frequency GNSS receivers are placed less than 20 km away from the reference station [55]. Depending on the solution of the carrier-phase ambiguity, the estimated positions have a quality parameter value: DGNSS, FLOAT, or FIXED (ordered from less to more accurate).

In the proposed UWB-based IPS, GNSS-RTK is used for the self-location of the IPS reference nodes (i.e., the UWB anchors). Usually, when performed manually, the positioning of reference devices is prone to become ill-posed and the performance and reliability of the positioning algorithms are inevitably influenced [56,57]. In our case, GNSS-RTK not only permits a fast and precise deployment of the UWB anchor network [58], but could even extend global positioning into an indoor scenario if applying the corresponding coordinate conversions.

### 3.3. Ranging Obtainment

As stated in the IEEE 802.15.4 standard, the medium access in a UWB network can be scheduled, so that devices are provided with slots to send and receive messages at known time intervals. Synchronization is achieved by means of periodic beacons, which are sent by the anchor acting as *master* (and, if needed, retransmitted by the anchors acting as *relays*), according to the value of the synchronization period ($T_{s}$). The time between two consecutive beacons makes up a superframe, as can be observed in Figure 3.

The proposed superframe contains $n_{\mathrm{TDMA}}$ slots of $t_{\mathrm{TDMA}}$ duration. Within each TDMA slot, a single network tag is entitled by the *master* anchor to send a UWB broadcast message during the so-called broadcast slot. Next, the tag will receive a set of consecutive responses during the subsequent *k* node slots, with *k* being the number of network anchors. Note that each node slot is allocated by the *master* to a single anchor. In consequence, after each TDMA slot, the tag should have received *k* UWB frames, one per available anchor. Lastly, a tag can be allocated with more than one TDMA slot per superframe.

The previously described UWB frame exchange allows us to extend the single-sided two-way ranging (SS-TWR) localization technique up to *k* anchors only using k+1 messages instead of 2·k messages, leading to a notable energy reduction [60]. According to the SS-TWR description [61], the broadcast message sent by the tag in our scheme would correspond to an SS-TWR request, whereas each anchor response would correspond to an SS-TWR reply. Then, as the tag knows the start time of each node slot and its corresponding allocated anchor, it is able to compute *k* ToFs, which will in turn be used to estimate *k* distances (*rangings*).

In case there is more than one tag to be tracked at the same time, the available $n_{\mathrm{TDMA}}$ slots must be distributed among the different tags. In fact, proper sizing of the main parameters of the superframe structure (namely, $T_{s}$, $n_{\mathrm{TDMA}}$, $t_{\mathrm{TDMA}}$, and *k*) is crucial, as there exists an important trade-off between scalability and accuracy. For example, for a given position update period (*T*), the more tags are tracked, the fewer computed rangings will feed the multilateration algorithm, likely impacting the positioning accuracy.

### 3.4. Tag-Positioning Algorithm

The multilateration algorithm is responsible for determining a stationary or moving device’s position based on the gathered rangings to each available anchor. In the proposed UWB-based IPS, this algorithm is based on the Gauss–Newton optimization method, which uses a polynomial function to approximate a given function value until a certain precision is reached. This method is well-suited for solving (small residual) non-linear problems and is most commonly applied in positioning applications [62].

Under the approach presented here, the multilateration algorithm is embedded in the tag, so that it can compute a new sample of its own position as long as two conditions are fulfilled:The number and position of the network anchors are well-known by the tag.The tag has received one ranging value from at least three different well-known anchors during a TDMA slot.

The first requirement is based on the dictionary of anchors’ positions, which is hosted in the command and control center and downloaded by the tag once it is powered on or if any modification occurs. The second one is accomplished by the continuous exchange of UWB frames between the tag and the available anchors according to the SS-TWR localization technique.

Lastly, all position samples computed over the *T* period with the Gauss–Newton optimization method are averaged to reduce the impact of outliers caused by the dynamism of the wireless channel, potential interference, and/or the movement of the tag itself. The resulting averaged position is sent to the command and control center.

### 3.5. Command and Control Center

The main role of the command and control center in the context of the UWB-based IPS is to support its operation by storing the information related to the network’s infrastructure (i.e., an updated dictionary of anchors’ availability and position), so that it can be downloaded by the tag when necessary. In addition, it also receives the positions of the FRs computed by the tags, stores them in a database, and enables their visualization in real time.

It is worth noting here that the use of cloud systems or even Internet access may not be fully available in disaster-affected areas due to power outages or unstable communication links. For that reason, when facing a real ER operation it would be highly advisable to move the command and control center as close to the disaster-affected area as possible. A dedicated emergency van is a very convenient option for this purpose, as it can quickly approach the incident area and provide FRs with the required services [63].

## 4. Implementation

### 4.1. Testbed Location

The performance of the UWB-based IPS was extensively assessed in a testbed deployed into and around a two-storey rectangular (20.11 × 12.88 × 6.20 m) building located in the south of the municipality of Egaleo within the Athens metropolitan area, with a total built area amounting to roughly 520.00 m${}^{2}$ (see Figure 4a). This building was later employed as one of the pilot scenarios of the RESPOND-A EU project, aimed to provide *next-generation equipment tools and mission-critical strategies for FRs* [64].

The single main entrance of the building is located on the ground floor. The material of the door is metallic, which totally covers the entrance. As for the façade, it is made up of concrete, whose thickness is 30 cm. The two floors are connected with stairs, share 4 columns, and have plenty of windows surrounding them (see Figure 5). As for the interior of the building, there are no walls, except for those of plasterboard that create two little rooms. However, the ground floor has plenty of tables and chairs, as well as scattered appliances and objects. None of this furniture exceeds 1.5 m in height.

A 3D model of the testbed building, whose general view can be found in Figure 4b, was designed to better understand potential UWB coverage issues, select the best position of anchors to maximize the network coverage area, and evaluate the accuracy and performance of the localization system on a visual basis.

### 4.2. Deployment

A set of 6 anchors was deployed on the outside of the testbed building with a twofold objective: (1) ensuring that the beacons emitted by the *master* are received by all the other anchors (whether directly or retransmitted by the *relays*) and (2) reaching full UWB coverage inside the building. To achieve this, apart from the *master*, two anchors needed to be set as *relays*, while the rest acted as *slaves* (see Figure 6a). Note that a *relay* is just an anchor that also retransmits beacons coming from the *master*.

To facilitate the propagation of UWB signals not only on the ground floor of the building but also on the first floor, the anchors were mounted on tripods at a height of 2–3 m and their antennas always pointed at the center of the building. Such a height of the deployed infrastructure ensured visibility among the anchors, thus contributing to accelerate UWB network establishment and avoid synchronization problems. Nevertheless, the lack of height diversity limited tag positioning to X-Y coordinates. Figure 6b,c show the anchor deployment on the front and back side of the building, respectively.

As a rule of thumb, there is a series of recommendations to be applied in the anchor deployment of the proposed UWB-based IPS:The UWB network infrastructure shall consist of at least 4 anchors (though multilateration can be achieved with only 3 elements, the extra one adds redundancy to the positioning system).The deployed anchors shall encircle the target building.The distance between two adjacent anchors shall be less than 30 m in LOS conditions.If the *master* anchor is further than 30 m from an anchor or it has NLOS conditions, *relays* shall be employed to ensure that the latter is properly receiving synchronization beacons.The anchors shall be placed at a height greater than 2 m to facilitate UWB signal propagation inside the building.Three consecutive anchors shall never be placed aligned in the same axis, because the multilateration algorithm could produce duplicate mirror positions or experience convergence issues.

As for the GNSS-RTK self-location system, apart from the satellites and the GNSS ground-based reference station(s), the on-field deployed infrastructure comprises a single portable GNSS-RTK module, which minimizes the total cost. Nevertheless, there is scope to embed this module into each operating anchor and create a more compact device in future versions of this IPS.

The GNSS-RTK module should be initially powered on for about 5 min in order to configure itself, that is, start detecting GNSS signals and receiving RTK corrections. Once the anchors are deployed, the GNSS-RTK module is connected to them consecutively. Each time the GNSS-RTK module is connected to an anchor, the latter sends its corresponding ID to the module and, after that, the self-location system starts.

From that moment on, the GNSS-RTK module stores positions for one minute and then evaluates if they are accurate enough according to a predefined threshold. If so, the positioning is considered successful, so the module can be disconnected from the current anchor and moved to another one. Otherwise, the GNSS-RTK module should be disconnected and reconnected to the same anchor to restart the self-location system.

### 4.3. Hardware

Both the anchor and the tag of the proposed UWB-based IPS consist of the same hardware elements:The **communication module** is a Qorvo DWM1001-DEV module development board [65], which integrates a DWM1001C UWB transceiver module, a Bluetooth antenna, all RF circuitry, a Nordic Semiconductor nRF52832 SoC, and a motion sensor.The **processing module** is an Espressif ESP-WROOM-32s development board [66], which includes a dual-core MCU and a WiFi antenna.The **power supply** is provided by a 2600 mAh Li-Ion battery.

The communication module is connected to the processing module through a UART interface, so that each new computed ranging is timely sent to the processing module. As for the power supply, the battery is directly connected to the communication module, from which energy is delivered to the rest of components. The whole set is laid on a 3D-printed support structure that enables easy mounting on a tripod or on a helmet (see Figure 7).

For its part, the GNSS-RTK module consists of the following elements:**Ardusimple simpleRTK2B budget board** [67]: a standalone board that allows one to evaluate dual-band GNSS-RTK technology. It is based on a u-blox ZED-F9P module and is fully compatible with Arduino and STM32 Nucleo platforms as a shield.**u-blox ANN-MB-00 L1/L2 multi-band antenna** [68]: a high-precision RTK multiband external GNSS antenna with 5 m cable and an SMA connector.**Ardusimple 4G NTRIP master** [67]: a radio module including cellular connectivity and the necessary software to connect the simpleRTK2B board to a GNSS real-time correction service. To validate the presented solution without deploying additional GNSS-RTK infrastructure, the NTRIP protocol [69] was used to receive corrections from the EUREF-IP service [70].**Raspberry Pi 4**: theprocessing unit of the whole GNSS-RTK module.

All the elements are connected to the Raspberry Pi, which initiates all the processes and computes the final GNSS-RTK-based position. In addition, the module incorporates an external LED with 4 colors that identifies all the stages of the anchor self-location system. This indicator eases the deployment of the GNSS-RTK self-location system performed by FRs in the field. Figure 8 shows the GNSS-RTK module, identifying all the aforementioned components.

### 4.4. Software

The software environment used to program the communication module of both the tag and the anchors is based on the *Apache Mynewt* operating system [71], a modular real-time operating system (RTOS) for connected IoT devices. In particular, a distribution of the manufacturer itself has been employed to manage the core UWB functions, such as those belonging to the medium access control (MAC) layer or the ranging computation process itself [72].

The processing module relies on the ESP-IDF development framework [73] as the basis of its main operations. In addition, FreeRTOS [74] is used to provide the processing module with multitasking capabilities, which are especially useful here to manage two continuous and simultaneous processes: (1) the ranging reception from the communication module and (2) the computation of positions by means of the multilateration algorithm.

The continuous flow of rangings is first introduced into a pre-filtering process that discards erroneous or out-of-bound values. While the former are easy to detect due to a special code stamped by the communication module, the latter are identified if they surpass a threshold based on the maximum possible distance inside the outer polygon created by the deployed anchors.

As for the multilateration process, an adaptation of the Gauss–Newton method implementation proposed in [75] has been employed. In short, the optimization function of this implementation receives a matrix with the positions of the anchors, a pointer to the distances to each anchor (i.e., the rangings), another pointer to the initial positioning values (which will be finally filled with the new position estimations), and lastly, the number of equations to solve.

There are two parameters with a great impact on the Gauss–Newton method behaviour and performance: the maximum number of expected iterations ($i_{\max}$) and the targeted precision in meters ($\hat{p}$). More specifically, the greater the $i_{\max}$ value and the lower the $\hat{p}$ value, the higher the accuracy of the positioning that will be obtained. In that case, though, more computing power and/or time will be needed to estimate a new position.

A cloud platform was also deployed to enable the visualization and facilitate the performance evaluation of the UWB-based IPS in real time. Its main elements are a webserver providing a REST API to receive the positioning information coming from the tag and an *InfluxDB* database to store that information. Communication between the tag and the cloud platform was established by means of a websocket accessed by the tag through its WiFi module.

For its part, the GNSS-RTK module processing unit (i.e., the Raspberry Pi) implements an automatic Python service that initiates at bootstart. It detects each time an anchor is connected to the Raspberry Pi and then starts the self-location routine. It also communicates with the processing module of the anchor through the serial communication interface. Once the receiving time of the GNSS-RTK information has finished, it processes the positioning data, averages the received GNSS-RTK samples to obtain the final position, and sends it to the cloud via WiFi.

## 5. Performance Evaluation

The proposed UWB-based IPS was subjected to different tests in the aforementioned building and surroundings to evaluate not only its performance but also its suitability to be employed in ER operations. This section compiles the main obtained results according to three categories: anchor self-location, static tag localization, and tag tracking.

### 5.1. Anchor Self-Location

The anchor self-location tests were performed in order to study the precision and accuracy of the proposed GNSS-RTK-based solution. For this study, the GNSS antenna of the GNSS-RTK module was placed on top of each anchor, analyzing a total of six locations.

A first analysis was performed on the dispersion of the position samples obtained during the localization phase. Figure 9 shows the temporal evolution of the received GNSS-RTK positions in each considered anchor location. For most of the locations, the system rapidly converged to a precise position. However, there were two anchors (namely, #8828 and #1A25) that presented a more disperse behavior.

Next, Table 2 presents the quantitative study of the obtained position samples, including the degree of FIXED level achievement, the mean of the self-estimated horizontal accuracy error ($\bar{hAcc}$), which provides a level of positioning uncertainty, and the computed standard deviation for both X and Y coordinates ($\sigma_{x}$ and $\sigma_{y}$, respectively). Both the quality parameter value related to the solution of the GNSS carrier-phase ambiguity (DGNSS, FLOAT, or FIXED) and the hAcc value were directly extracted from the u-blox ZED-F9P module. More information can be found in its interface description [76].

The obtained values corroborate the graphical representation in Figure 9, where all the obtained position samples in each considered location converge to a stable point. In addition, the two anchors that showed a bigger dispersion also had slightly worse $\bar{hAcc}$ values and some of their samples did not even reach the FIXED ambiguities. In this matter, the area where these anchors were placed did not have the best conditions, as the GNSS ground-based reference station sending the corrections was located further than recommended. Even so, this scenario has been considered convenient to analyze the performance of the system under realistic/non-ideal conditions.

The evaluation of the positioning accuracy was conducted by analyzing the distance error between anchors. The study was carried out on the final position obtained once the self-location process was completed. Typically, the accuracy is calculated by comparing the computed position to the real one. However, since there was no alternative location method to determine the exact position of the anchors, the calculation was carried out indirectly with distances.

In the presented testbed, the real distances between the anchors were calculated manually with different mechanisms: a tape measure and a laser meter. Although these measurement campaigns are not error-free, the resulting values can be effectively used as a benchmark and compared to those obtained from GNSS-RTK. The difference between both value sets, compiled in Table 3, can be considered as the GNSS-RTK error when computing distances between anchors and offers an idea of the resulting accuracy. Particularly, most of the errors remain below 30 cm and only a few surpass this threshold. Again, the anchors that show higher distance errors correspond to those that also presented worse precision (i.e., #8828 and #1A25).

The time devoted to the deployment and activation of the IPS ($T_{\mathrm{IPS}}$) may be critical in an ER operation. In the IPS presented here, two processes can be run in parallel: the deployment and calibration of the GNSS-RTK self-location system (with a duration of $T_{\mathrm{GNSS}-\mathrm{RTK}}$), and the synchronization of the UWB network (with a duration of $T_{\mathrm{UWB}}$). Generally, $T_{\mathrm{UWB}}\ll T_{\mathrm{GNSS}-\mathrm{RTK}}$, so that $T_{\mathrm{IPS}}\equiv T_{\mathrm{GNSS}-\mathrm{RTK}}$.

Equation (Equation 1) details the different elements of $T_{\mathrm{GNSS}-\mathrm{RTK}}$, where $T_{start}$ corresponds to the GNSS-RTK initial startup time, *k* to the number of anchors to deploy, $T_{mov}$ to the time devoted to move the GNSS-RTK module from one anchor to another, and $T_{pos}$ to the GNSS-RTK positioning time. In the presented testbed with k= 6 anchors these timings correspond to $T_{start}=300$ s, $T_{mov}=30$ s, and $T_{pos}=60$ s. In consequence, $T_{\mathrm{IPS}}\geq 840$ s (i.e., ≥14 min).
(1)$$T_{\mathrm{IPS}}\equiv T_{\mathrm{GNSS}-\mathrm{RTK}}\geq T_{start}+k\cdot \left(T_{mov}+T_{pos}\right)$$

An anchor can be deployed by just one person. However, if all the anchors had their own embedded GNSS-RTK module and the deployment could be performed by a team of *n* FRs (being 1≤n≤k), $T_{\mathrm{IPS}}$ could be greatly reduced (see Equation (Equation 2)). Specifically, for n= 2, the testbed presented herein could have been deployed in 450 s (i.e., 7.5 min).
(2)$$T_{\mathrm{IPS}}\equiv T_{\mathrm{GNSS}-\mathrm{RTK}}\geq T_{start}+\left\lceil \frac{k}{n}\right\rceil \cdot T_{mov}+T_{pos}$$

Furthermore, the potential use of a UAV constellation, where each UAV would act as an anchor supporting GNSS-RTK self-positioning capabilities, would not only reduce $T_{\mathrm{IPS}}$ (see Equation (Equation 3)), but also simplify the whole IPS infrastructure-deployment process. Moreover, this option would improve the flexibility and dynamism of the infrastructure, since it could be redeployed and adapted to a changing environment in a simple manner.
(3)$$T_{\mathrm{IPS}}\equiv T_{\mathrm{GNSS}-\mathrm{RTK}}\geq T_{start}+T_{mov\;\left(\mathrm{UAV}\right)}+T_{pos}$$

### 5.2. Static Tag Localization

A set of 18 well-known positions (from now on called *puntX*) were defined to measure the performance of the UWB-based IPS when locating static objects. As can be observed in Figure 10, these positions aimed to represent the whole building, with eight of them placed on the ground floor and ten on the first floor. During the test, a tag was placed in each position on top of a tripod at a height of 1.80 m for less than 2 min. Table 4 compiles the rest of the configuration parameters of the proposed UWB-based IPS.

As a first step of the static tag localization analysis, the SS-TWR procedure execution and the accuracy of the obtained rangings was evaluated. (A dataset with all the obtained rangings from this test is available at https://bitbucket.i2cat.net/users/toni_adame/repos/uwb-rangings, accessed on 21 April 2023). As shown in Figure 11a, in 96 of the 108 considered tag–anchor pairs (i.e., 18 tag positions × 6 anchors), the tag properly processes more than 90% of the expected rangings. In such cases, the few lost rangings are due to brief communication or processing issues. However, in some positions the tag experiences a permanent lack of UWB coverage from one or more anchors. This is particularly significant in *puntK*, where only three anchors are *visible* more than 90% of time and the rest are usually *unreachable*.

The mean absolute ranging error with respect to the actual tag–anchor distance is depicted in Figure 11b. The mean error is below 1 m in 80.56% of all considered tag–anchor pairs and achieves its maximum value (2.65 m) in the *puntM*–*anchor #1a25* pair. At this point it is worth noting that the achieved performance is far from the optimum centimeter-level benchmark offered by UWB due to the strict NLOS character of the current testbed and the range overestimation introduced by the high distance between devices [77].

In particular, some of the greatest error values correspond to pairs with a low rate of received rangings, which confirms the instability of the wireless link in such cases. Another remarkable trend is that the error values greater than 1 m mainly belong to positions on the first floor of the building. The effect of the ground-floor ceiling, which acts as an additional obstacle hindering UWB signal propagation, is behind this behavior.

The distance between the well-known position and the anchor does not constitute a major contributor to the ranging error, since the computed correlation coefficient between these two metrics accounts for 0.15 (here computed as the Pearson correlation coefficient, which is defined as $\rho \left(A,B\right)=\frac{1}{N-1}\sum_{i=1}^{N}\left(\frac{A_{i}-\mu_{A}}{\sigma_{A}}\right)\left(\frac{B_{i}-\mu_{B}}{\sigma_{B}}\right)$, where $\mu_{A}$ and $\sigma_{A}$ are the mean and standard deviation of *A*, respectively, and $\mu_{B}$ and $\sigma_{B}$ are the mean and standard deviation of *B*, respectively). On the other hand, the dispersion of the computed rangings shown in Figure 11c does slightly correlate with the ranging error, with a computed coefficient of 0.42.

In any case, the obtained rangings fed the multilateration algorithm, which estimated the tag position every T∼2 s. Figure 10 shows the projection on the building map, per well-known point, of all obtained estimation samples and their average value. At first sight, the cloud of estimations is always located close to the actual tag position with a reduced number of outliers, with those mainly located on the first floor.

A quantitative analysis of the accuracy and precision of these estimations is provided in Figure 12. The first metric of interest corresponds to the average 2D distance error in localization, which is always below 1 m, but shows a variable behavior depending on the tag’s position. Apart from the aforementioned strict NLOS conditions between the tag and the available anchors, there are two other factors that explain this performance variability:**Number of detected anchors:** Similarly to satellite positioning, dilution of precision (DOP) describes the relationship between the measurement error and position determination [62]. Therefore, the more observations used (in our case, the more detected anchors), the smaller the DOP values and, hence, the smaller the solution error.**Geometry delimited by detected anchors:** Another parameter, called the geometric dilution of precision (GDOP), is a dimensionless value that measures the effect of the network geometry on the positioning solution [78]. In short, if a square pyramid is formed by lines joining the anchors with the tag at the tip of the pyramid, the larger the total volume, the better the value of GDOP [79]. Consequently, a high share of aligned anchors could penalize the positioning performance.

The better behavior of the IPS in some positions is likely to be caused by (1) the avoidance of the effect of the ground-floor ceiling and other structural elements such as the stairs, which hinder the UWB signal propagation, (2) the reduced number of obstacles across the UWB signal path to the anchors, and (3) the greater number of nearby available anchors, which tend to provide more accurate rangings.

As for the standard deviation of the computed data, it can be observed that the localization precision is, in general, lower in positions of the first floor (with the exception of *puntG*). Again, the signal propagation variability introduced by the higher number of obstacles leads to a certain dispersion in the position estimation. Still, there are some exceptions, such as *puntN*, where the localization estimations are highly precise but not that accurate (see also Figure 10b), because the average error corresponds to 0.53 m.

In general, the obtained 2D accuracy values are not lower bounds; in fact, this metric can be improved by selecting a subset of anchors involved in the multilateration algorithm. Indeed, it is well-known that the selection of certain anchors from a whole set can affect the localization accuracy significantly [80]. While it is true that this topic falls outside of the scope of the present work, it has been largely discussed in the existing UWB literature [81,82,83,84,85] and will be the subject of most of our next research efforts. More specifically, it seems clear that the implementation of an anchor-selection method that takes into consideration the rangings’ faithfulness during each *T* period and the geometry of the deployed infrastructure would notably enhance the accuracy of the IPS presented herein.

### 5.3. Tag Tracking

The UWB-based IPS was also tested on a mobile object. The tracking of an FR during an ER operation was simulated according to the guidelines provided by FRs from the City of Barcelona’s Fire Department. Specifically, a user carried a tag placed on a helmet while performing a predefined path throughout the target building. The speed *v* at which the user walked was assumed to be slow (i.e., v< 1 m/s), as it was intended to reproduce the movement of an FR during an ER operation, where heavy equipment may be carried and caution is necessary when approaching or being inside a disaster-affected area. Different paths were designed with the aim of covering the different rooms of the building and evaluate the performance of the tracking system in all those spaces. Again, Table 4 compiles the rest of the configuration parameters.

Figure 13 shows the expected and the computed tracking of *pathA* on the ground floor of the target building. This path simulates the movement performed by an FR entering the building, looking for victims in that floor, and exiting the building. In addition, it passes through six well-known reference points. As can be observed, the path followed by the user is reproduced with high faithfulness, even in the narrow aisles of the *north* side of the building. In general, the estimated positions are rarely placed on obstacles or far from the actual aisle.

Similarly, *pathB* simulates the path followed by a FR looking for victims after having accessed the first floor through the stairs (see Figure 14). All reference points from this floor were also included in the path. While it is true that no aisles are available to assess the accuracy, the path followed between two consecutive reference positions is always straight except for the *puntI* →*puntJ* section. Specifically, some deviation is detected around *puntI*, where the estimated positions are roughly 2 m closer to the wall than they should be. As for the *puntQ*→ *puntP* section, it was explicitly performed in two different straight movements due to the presence of some objects on the floor.

Lastly, the positioning system was also evaluated outdoors in pathC. More explicitly, a user walked through the building’s external shape in a counterclockwise direction, as shown in Figure 15, with the tag always confined inside the outer polygon created by the deployed anchors. Therefore, this test proves the continuity of the proposed UWB-based IPS while transitioning to a potentially available absolute positioning system (e.g., based on GPS technology) when the FR goes into the open air.

The shape of the building can be easily inferred from the obtained tracking, with two remarkable exceptions highlighted in yellow color in Figure 15b: (1) an erroneous estimated position that places the tag closer to anchor *#8828* than expected and (2) an area with some position divergence between anchors *#5BA0* and *#3EB*. (The area between anchors *#2CDB* and *#1A25* was not accessible during the test execution.) A deeper analysis of the obtained rangings may explain the bad performance of the positioning system in the two aforementioned cases:In the first case, the tag has computed rangings from all anchors except for *#5BA0*. In addition, the ranging values from anchors on the opposite side of the building (i.e., *#1A25* and *#3EB*) are larger than the real distance, moving the final position estimation far from the building’s shape.Similarly, in the second case there is an important lack of rangings coming from anchors on the opposite side of the building (i.e., *#2CDB*, *#8828*, and, to a lesser extent, *#1A40*). Especially remarkable is the effect of anchor *#8828*, whose few computed ranging values were greater than the actual distance, leading to the observed distortion in the estimated position.

To sum up, in both the analyzed areas the combination of high distance (> 20 m) and obstacles (at least two outer walls) between the tag and an anchor led to the reception of untrustworthy ranging values or even the lack of them. In consequence, tag positioning may not be accurate enough in such areas, and thus increasing the anchor density and/or implementing dynamic filtering techniques to remove information from unreliable anchors may be required.

## 6. Conclusions

The critical character of actions performed by FRs during ER operations requires that any technological solution aimed to support them be non-intrusive, effective, quick to deploy, as well as easy to manage and use. Providing command and control centers with an IPS of such features, coupled with simultaneous awareness of incident risks, would strengthen disaster preparedness for a safe and effective response in such operations.

Commercial and well-established infrastructure-based IPSs have already been proven suitable for a heterogeneous set of use cases (e.g., healthcare, industry, and retail). However, they may be rendered inoperable during extraordinary situations such as those requiring ER operations and induce substantial costs in terms of both setup and maintenance. As an alternative, the UWB-based IPS implemented herein fulfills the performance requirements of ER operations while being easily and quickly deployed, operated, and dismantled after its use.

One of the most time-consuming tasks for IPSs without pre-existing infrastructure is the initial deployment and setup of reference devices. To accelerate this process, the presented solution implements an automated, worldwide-available, and easy-to-deploy self-location system based on GNSS-RTK. The positioning of each reference device is accomplished in one minute, reaching centimeter-level accuracy in the vast majority of the obtained position samples. Moreover, the resulting absolute reference system created by the UWB anchors could be further employed to locate indoor assets under global coordinates, even under non-ideal GNSS-RTK conditions.

The obtained results in a testbed building with strict NLOS conditions show how the UWB-based IPS is able to locate a static tag in a wide set of positions inside the building with less than 1 m of error. Furthermore, real-time tag tracking tests allowed us to reconstruct the path followed by a user simulating the movements of a FR during an ER operation in a timely manner. In fact, the designed low-cost UWB tag mounted on the top of a helmet was able to effectively compute and notify its own position not only inside the target building but also in its closest surroundings.

Still, there is wide room to further increase the accuracy of the UWB-based IPS in NLOS conditions. Three potential research lines are here clearly identified: (1) to incorporate UWB communication modules supporting the IEEE 802.15.4z standard, which includes a set of enhancements to ensure reliable and safe ranging [86]; (2) to dynamically detect and remove ranging values from unstable and/or unreliable tag–anchor links before they are introduced into the multilateration algorithm; and (3) to implement a sensor fusion system in the tag, integrating both speed and acceleration measurements from an IMU, which could provide positioning continuity in areas with low or no UWB signal coverage.

Furthermore, the particular requirements and constraints of ER operations lead to the addressing of the following challenges: (1) the incorporation of unmanned aerial vehicles (UAVs) and automated guided vehicles (AGVs) into the IPS infrastructure to create UWB mobile networks able to follow the operation, achieve full multistory positioning, and sketch digital maps [87,88]; (2) the assessment of the benefits associated with the looser transmission limitations ruling over LAES-based UWB equipment; (3) the miniaturization and integration of a lightweight tag into already-existing FR equipment (e.g., a firefighter’s helmet); and (4) communication redundancy between the tag and the command and control center.

## Figures and Tables

**Figure 1 sensors-23-04193-f001:**
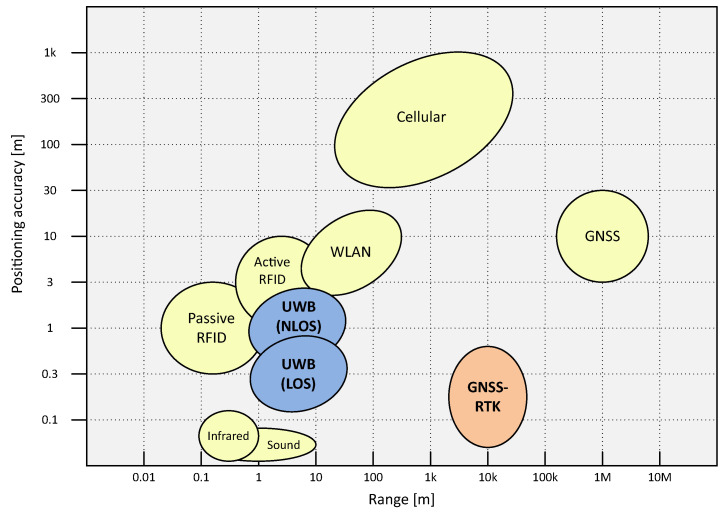
Feature comparison of the most important positioning technologies (based on [21]).

**Figure 2 sensors-23-04193-f002:**
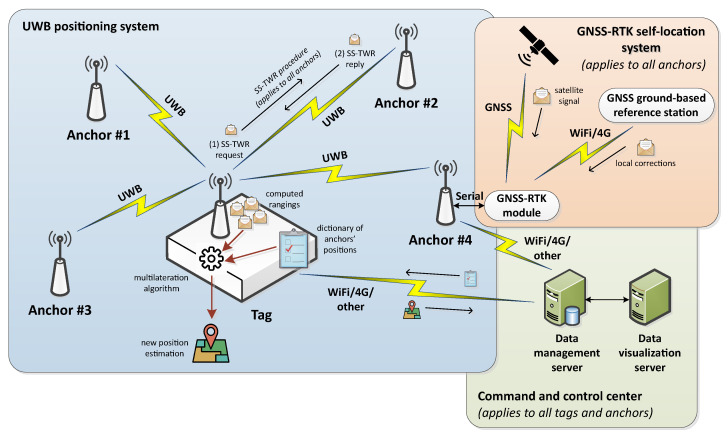
Architecture of the proposed UWB-based IPS.

**Figure 3 sensors-23-04193-f003:**
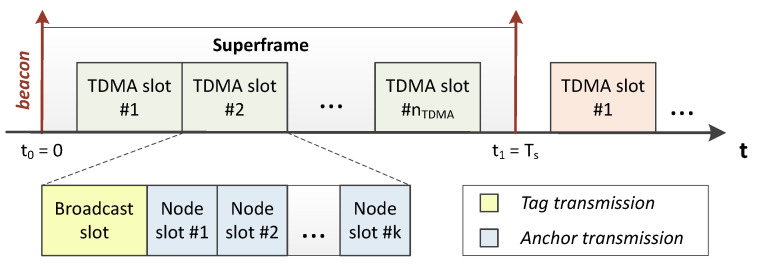
Superframe structure (simplified version of the diagram published in [59]).

**Figure 4 sensors-23-04193-f004:**
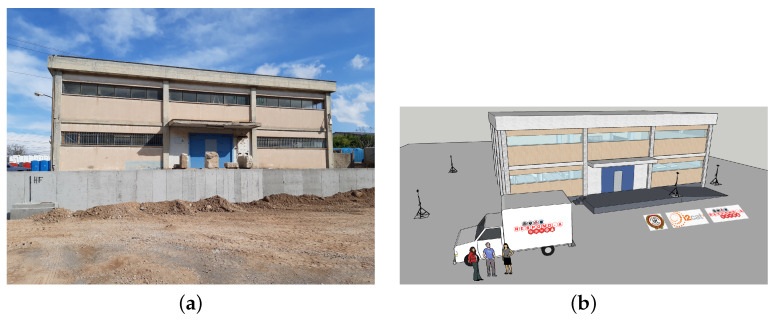
Testbed building. (**a**) Front view. (**b**) 3D model.

**Figure 5 sensors-23-04193-f005:**
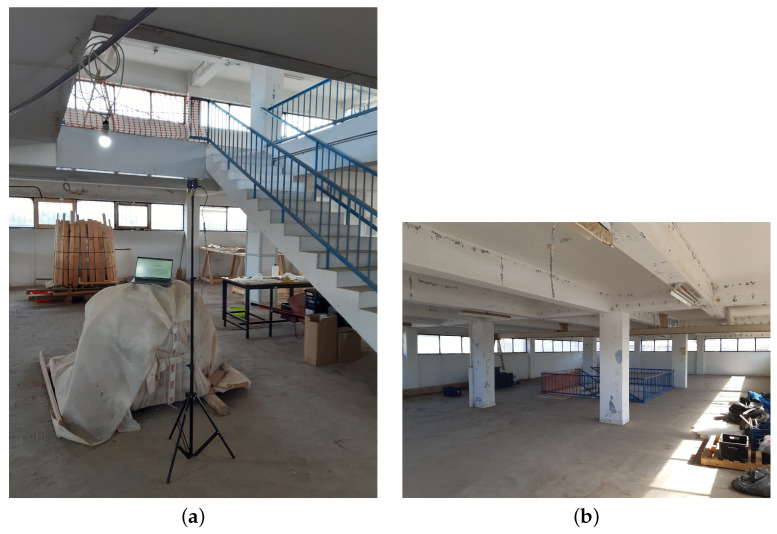
Interior of the building. (**a**) Ground floor. (**b**) First floor.

**Figure 6 sensors-23-04193-f006:**
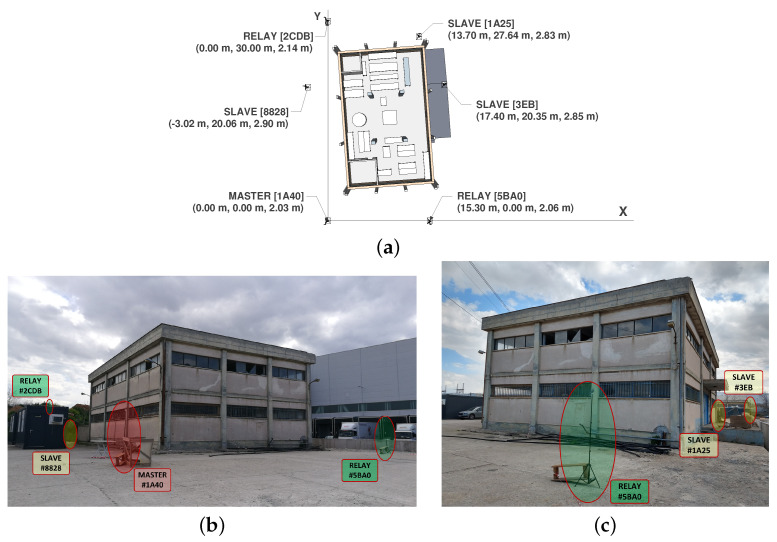
Anchor deployment. (**a**) Top view. (**b**) Front view of the building. (**c**) Back view of the building.

**Figure 7 sensors-23-04193-f007:**
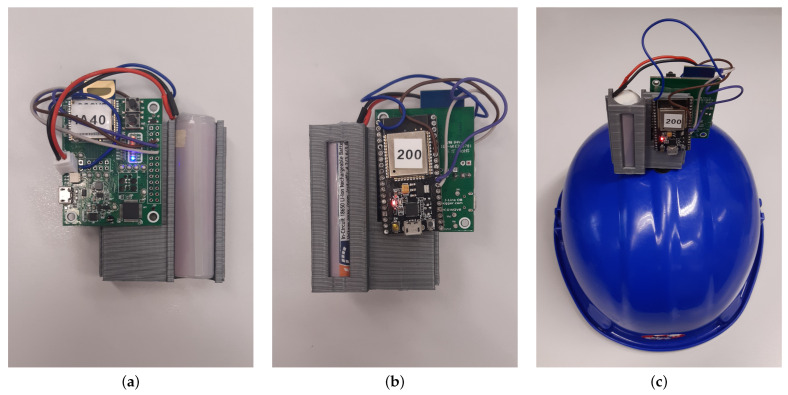
Designed UWB device (used either as an anchor or as a tag). (**a**) Front view. (**b**) Back view. (**c**) Mounted on a helmet.

**Figure 8 sensors-23-04193-f008:**
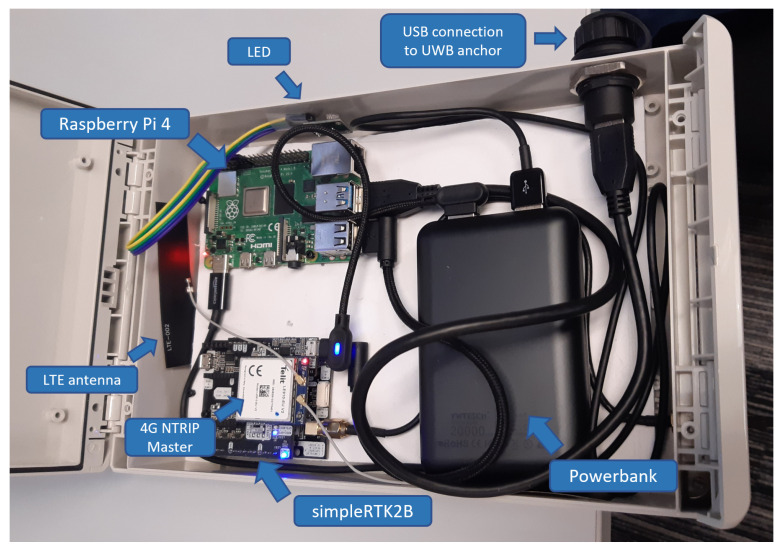
GNSS-RTK module.

**Figure 9 sensors-23-04193-f009:**
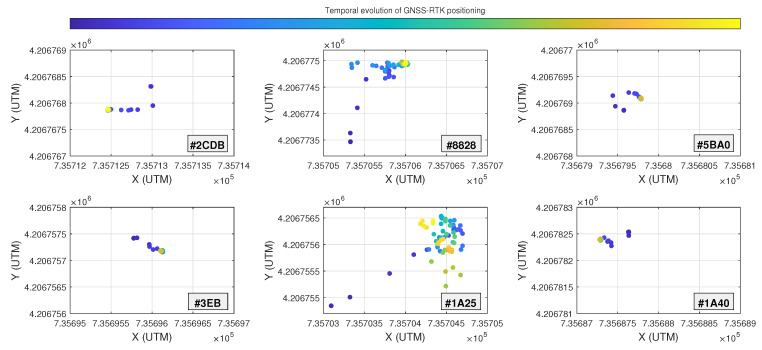
Dispersion of GNSS-RTK samples. All the subfigures plot an area of 2 × 2 m.

**Figure 10 sensors-23-04193-f010:**
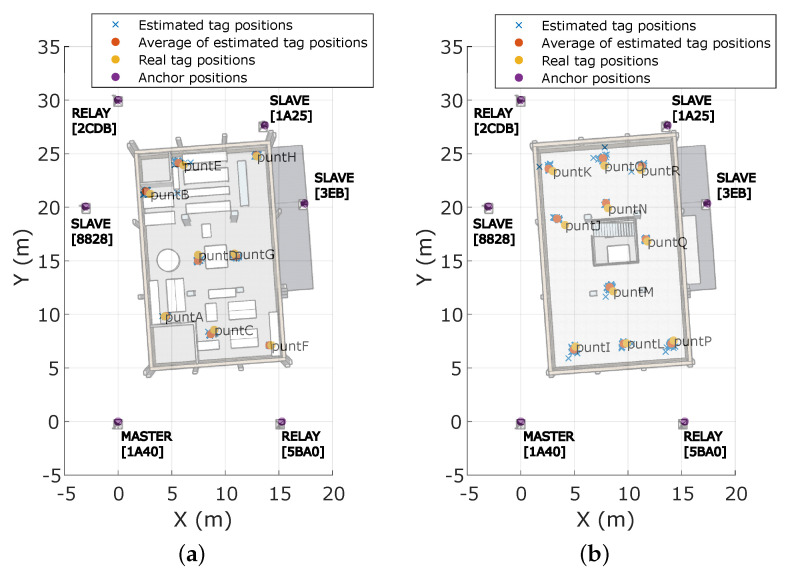
Static localization maps. (**a**) Ground floor. (**b**) First floor.

**Figure 11 sensors-23-04193-f011:**
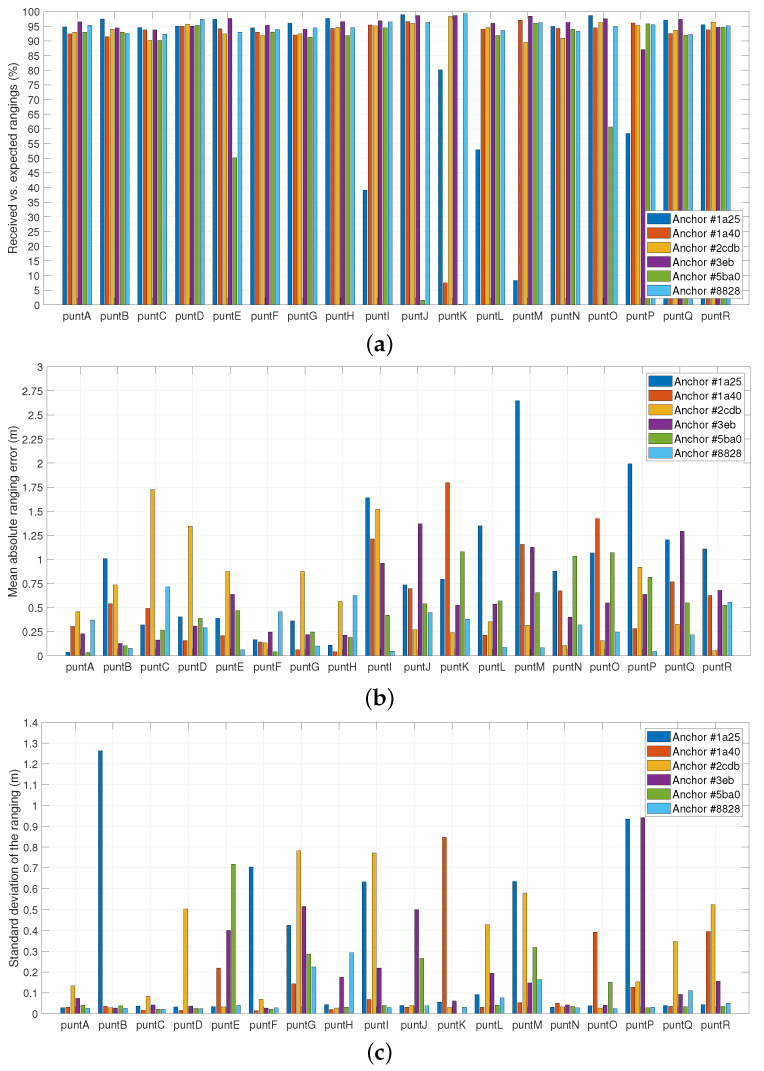
Statistics of computed rangings. (**a**) Received rangings. (**b**) Ranging error. (**c**) Ranging deviation.

**Figure 12 sensors-23-04193-f012:**
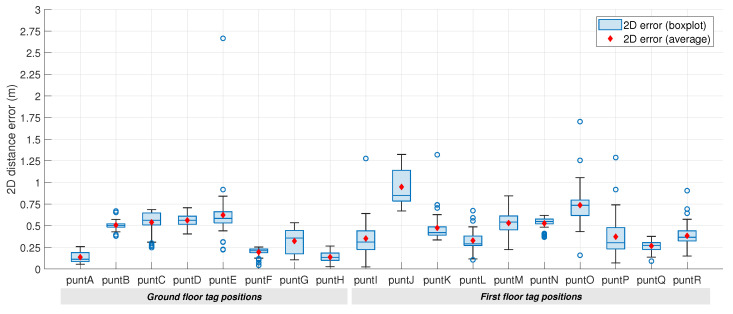
Static localization accuracy and precision.

**Figure 13 sensors-23-04193-f013:**
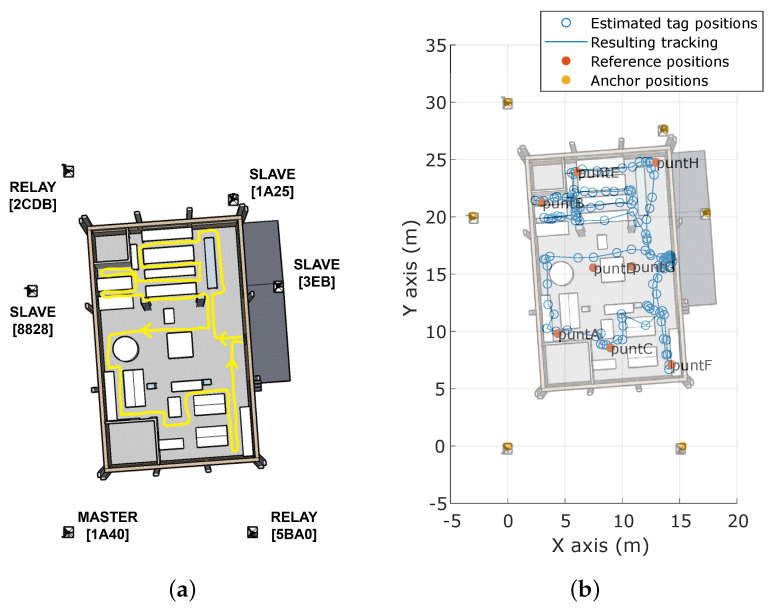
Tracking map of *pathA* on the ground floor. (**a**) Executed path. (**b**) Computed tracking.

**Figure 14 sensors-23-04193-f014:**
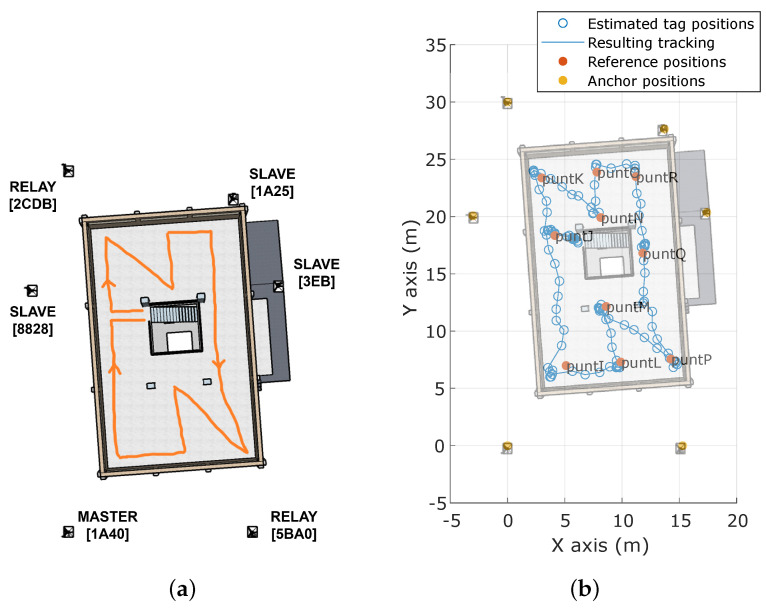
Tracking map of *pathB* on the first floor. (**a**) Executed path. (**b**) Computed tracking.

**Figure 15 sensors-23-04193-f015:**
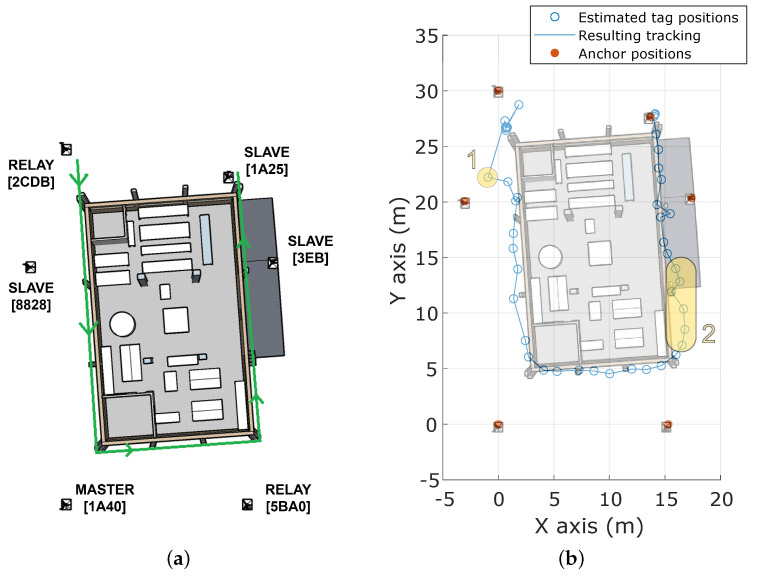
Tracking map of pathC outdoors. (**a**) Executed path. (**b**) Computed tracking.

**Table 1 sensors-23-04193-t001:** Comparison table of short-range, RF-based IPS technologies ([17,20,24], and authors).

Technology	Frequency	Potential Inferference	Operational Range	Positioning Parameter(s)	Positioning Accuracy
Active RFID	UHF and 2.4 GHz *	Medium/high	15 m	RSSI	2–3 m
Passive RFID	LF, HF, and UHF *	Low/medium	10 cm–2 m	RSSI	0.5–1 m
BLE	2.4 GHz	Medium/high	30 m	RSSI, TDoA	2–5 m
UWB	3.1–10.6 GHz	Very low	30 m	AoA, ToA, TDoA, TWR, PDoA	30–50 cm
WiFi	2.4/5 GHz	High	50 m	RSSI, AoA, ToA	2–5 m
ZigBee	2.4 GHz	Medium/high	100 m	RSSI	3–5 m

* LF: 125/134 kHz. HF: 13.56 MHz. UHF: 433/860–960 MHz.

**Table 2 sensors-23-04193-t002:** GNSS-RTK precision study.

Anchor ID	% of Samples with FIXED Ambiguities	$\bar{\mathrm{hAcc}}$ (cm)	$\sigma_{x}$ (cm)	$\sigma_{y}$ (cm)
**#1A40**	100	1.41	9.09	3.90
**#2CDB**	100	1.41	15.22	11.13
**#8828**	100	1.43	19.78	28.43
**#5BA0**	100	1.41	8.10	6.59
**#3EB**	100	1.41	8.94	6.43
**#1A25**	99	1.42	23.98	33.71

**Table 3 sensors-23-04193-t003:** Absolute distance error between anchors according to the GNSS-RTK self-location system (cm).

Anchor ID	#1A40	#2CDB	#8828	#5BA0	#3EB	#1A25
**#1A40**	-	7	29	2	9	39
**#2CDB**	7	-	21	0	19	47
**#8828**	29	21	-	44	24	33
**#5BA0**	2	0	44	-	26	8
**#3EB**	9	19	24	26	-	16
**#1A25**	39	47	33	8	16	-

**Table 4 sensors-23-04193-t004:** UWB-based IPS parameters.

UWB–PHY Parameters	Value
Channel number	3
Center frequency ($f_{c}$)	4492.8 MHz
Bandwidth (BW)	499.2 MHz
Pulse repetition frequency (PRF)	64 MHz
Preamble length	128 symbols
Data rate (*r*)	850 kbps
Preamble code ($c_{pr}$)	9
PAC size	8 symbols
Frame delimiter	Non-standard
PHR mode	Extended
CRC filter	Off
Transmission power	<−41.3 dBm/MHz
**UWB–MAC parameters**	**Value**
Synchronization period ($T_{s}$)	1.04 s
Number of TDMA slots ($n_{\mathrm{TDMA}}$)	10
Number of node slots (*k*)	6 (= number of anchors)
**Gauss–Newton algorithm parameters**	**Value**
Maximum number of expected iterations ($i_{\max}$)	1000
Targeted precision ($\hat{p}$)	0.001 m
**Application parameters**	**Value**
Position update period (*T*)	∼2 s

## Data Availability

The data presented in this study (specifically, those related to the obtained UWB rangings in the described testbed building) are openly available at https://bitbucket.i2cat.net/users/toni_adame/repos/uwb-rangings, accessed on 21 April 2023.

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
