# Peer review of "Fast Deployment of a UWB-Based IPS for Emergency Response Operations"

_sensors, 2023, doi:10.3390/s23094193_

Round 1

Reviewer 1 Report

Hello, The overall manuscript looks fine to me. Thank you so much.

I just have one concern and that is at:

Line 184-185: "UWB is not affected by interference from other communication devices or external noise due to its high bandwidth and signal modulation " This statement cannot be always valid. For example at UWB Ch5, it has very high interference with 6GHz band for wifi 7 application, even at Ch9 which is at 8GHz you can see coexistence issue with the 6GHz band.  

Reviewer 2 Report

This paper presents a UWB-based indoor positioning system (IPS) with anchors located by the GNSS-RTK technology. Specifically, it first employs the GNSS-RTK technology. to quickly, accurately, and safely deploy the anchors on outdoor sites, then it uses the UWB-based single-sided two-way ranging (SS-TWR) measurements to position the TAG.  Moreover, experiments have been conducted to demonstrate the performance of the proposed presented system. Yet, there are some problems in the paper: 

1.  In this paper, the presented indoor positioning system combines both the existing GNSS-RTK technology and the typical UWB-based indoor positioning technology. However, such a combination is trivial.

2.  As we knew, the UWB signal can provide accurate ranging estimation. However, the UWB-based single-sided two-way ranging (SS-TWR) measurements in the experiments is not such a case, please explain this.

3.  As shown in the simulation results, the positioning performance and the ranging performance depend on the TAG position, Please provide theoretical analysis on this dependence besides the simulation. results and the comments.

4.  Since the proposed positioning method is not compared with existing similar methods, the simulation results is not convincing. More simulation results comparing the proposed method with existing methods are needed. Moreover, the tracking performance is based on just one moving trajectory, the statistical tracking results would be more convincing.

Reviewer 3 Report

In this paper, the authors proposed a fast deployment of a UWB-based IPS for emergency response operations. The proposed scheme is based on a UWB-based IPS that relies on the global navigation satellite system-real-time kinematic (GNSS-RTK) technology to quickly, accurately, and safely deploy its required infrastructure on site. To evaluate the effectiveness of the proposed scheme, a set of tests was conducted on a two-story building and it was shown that an average accuracy of less than 1 meter for static targets and the ability to faithfully reproduce the path followed by a mobile target inside the building. As an additional comments, the positioning performance of the proposed scheme needs to be evaluated in higher building with several stories.The manuscript is well written and the contribution is solid, and especially the implementation in real building and the analysis of its performance provides good contribution to other researchers. It is good enough to be accepted and published.

Round 2

Reviewer 2 Report

All my concerns have been addressed by the authors. There is no more comments.